# Lip, Oral Cavity and Pharyngeal Cancer Burden in the European Union from 1990–2019 Using the 2019 Global Burden of Disease Study

**DOI:** 10.3390/ijerph19116532

**Published:** 2022-05-27

**Authors:** Aisling O’Sullivan, Zubair Kabir, Máiréad Harding

**Affiliations:** 1Ahern’s Dental Practice, Killorglin, Co. Kerry Bridge Place Dental Practice, Tralee, Co. Kerry, Cork University Dental School and Hospital, University College Cork (UCC), 200444 Cork, Ireland; aislingosullivan@ucc.ie; 2School of Public Health, Western Gateway Building, University College Cork (UCC), 200444 Cork, Ireland; 3Oral Health Services Research Centre, Cork University Dental School and Hospital, University College Cork (UCC), 200444 Cork, Ireland; m.harding@ucc.ie

**Keywords:** oral cancer, incidence, GBD, epidemiology, public health

## Abstract

***Background***—The recent burden of lip and oral cavity, nasopharynx and other pharynx cancer (LOCP) has not been specifically investigated in Europe. ***Methods***—In this descriptive epidemiological study, LOCP was categorised into lip and oral cavity cancer, nasopharynx cancer and other pharynx cancer, with European trends documented using the 2019 Global Burden of Disease (GBD). Summary statistics included deaths, age-standardised incidence rates (ASIR), mortality rates, YLLs (years of life lost), YLDs (years of life lived with disability) and DALYS (disability-adjusted life years). ***Results***—Lip and oral cavity cancer (LO) is the most dominant with the incidence decreasing from 6.2 new cases per 100,000 (95% UI: 6.1–6.4) in 1990 to 5.3 new cases per 100,000 (95% UI: 4.6–6.1). However, nasopharynx cancer (NP) and other pharynx cancer (OP) increased from 1 and 2.2 new cases per 100,000 in 1990 to 1.1 and 3.3 new cases per 100,000 in 2019, respectively. It was noted that LOCP YLLs is much higher than YLDs. In Europe, eastern European countries, specifically Hungary, have the highest burden of LOCP. When LOCP attributable to tobacco in Ireland was compared with the EU, the percentage decrease in OP DALYs attributable to tobacco is below the EU average, whereas the percentage decrease in LO attributable to tobacco in Ireland was above the EU average. ***Conclusions***—There has been a significant increase in ASIR in categories other pharynx and nasopharynx cancer since 1990, with significant geographic variations.

## 1. Introduction

Lip, oral cavity, nasopharynx and other pharynx cancer (LOCP) claimed the lives of approximately 35,000 persons in Europe in 2019 [1]. LOCP constitutes a significant European public health issue with social, physical and economic impacts [2]. Previous research has shown that eastern European countries have disproportionately high rates of LOCP [3]. A LOCP diagnosis can result in debilitating consequences, including oral resection, reconstruction and treatment-induced xerostomia [4]. Research carried out in the United Kingdom (UK) highlighted the average health service cost for LOCP surgical resection is almost EUR 24,000 per patient, with 32.3% of patients dying within five years [5]. LOCP risk factor prevention, along with early detection and diagnosis, can significantly reduce the LOCP mortality rate [6]. Risk factors include smoking, alcohol consumption, ultraviolet light and HPV [6]. The burden of LOCP, as opposed to measuring incidence and mortality rates, provides a comprehensive understanding of the impact of LOCP on the European population in terms of both premature mortality and disability [7]. The 2019 Global Burden of Disease study (GBD) offers a powerful resource of 359 diseases, including LOCP [1]. Led by the Institute for Health Metrics and Evaluation (IHME), 2019 GBD is the most extensive worldwide observational epidemiological study to date [8]. The metric DALYs—disability-adjusted life year (DALY)—is a single measure to quantify the burden of diseases, injuries, and risk factors. DALYs are based on years of life lost from premature death (YLL) and years of life lived in less than full health, otherwise termed YLD [9].

The aim of this research is to document trends of LOCP deaths, age-standardised incidence rate (ASIR), age-standardised mortality rate (ASMR), years of life lost (YLL), years of life lived with disability (YLD) and disability-adjusted life years (DALYs) using GBD data from 1990 to 2019 for European countries, to estimate the burden of LOCP in northern, southern, eastern and western Europe, using 2019 UN Statistics Division Classification to observe if any underlying patterns that suggest disparity exist and to compare LOCP DALYs attributable to tobacco use in Ireland and EU between 1990 and 2019, using Ireland as a stand-alone objective.

The results aim to assist EU policymakers to allocate region-specific risk factor preventative programmes, promote early detection of LOCP and highlight best practices for LOCP prevention and diagnosis.

## 2. Materials and Methods

In advance of undertaking this research, a comprehensive review of the literature was undertaken to identify previously published studies in a similar area. All data were obtained from the Global Health Data exchange (GHDx), which were compiled by the IHME. The reporting of the data estimation process followed the Guidelines for Accurate and Transparent Health Estimates Reporting (GATHER) for consistency and reproducibility. Data were gathered under appropriate GBD classifications B 1.1 lip and oral cavity cancer (LO), B 1.2 nasopharynx cancer (NP) B 1.3 other pharynx cancer (OP). Data were analysed for the years 1990 and 2019 to identify current LOCP trends following the introduction of public health measures in Europe, such as anti-smoking policies. In addition, no previous GBD publication has compared the EU LOCP trends in 1990 and 2019.

A combination of summary metrics was used to quantify the burden of LOCP including deaths, ASIR, ASMR, YLLs, YLD and DALYs. Years of life lost was selected as it measures the reduction in life expectancy due to LOCP. It is calculated as the number of deaths (n) x the standard life expectancy at age of death (L1) [9]. Years of life lived with disability represents the diminished quality of life experienced by an individual with injury or illness [9]. It is the number of new cases of a disease (I) x a disability weight (DW) x the average time a person lives with the disease before remission or death (L2) [9]. DALYs were chosen to comprehensively account for and compare the heath loss from fatal and non-fatal illnesses [9]. DALYS is the sum of YLL and YLD (DALYS = YLL + YLD) [9].

The following GBD age categories were included: age-standardised, 15–49 years, 50–69 years and 70+ years. Gender differences were analysed. Countries in the EU were included. For our study, the EU was defined as 28 member states in January 2019, i.e., Austria, Belgium, Bulgaria, Croatia, Cyprus, Czech Republic, Denmark, Estonia, Finland, France, Germany, Greece, Hungary, Ireland, Italy, Latvia, Lithuania, Luxembourg, Malta, Netherlands, Poland, Portugal, Romania, Slovakia, Slovenia, Spain, Sweden, U.K. Data were analysed as follows: overall trends, LOCP categories, GBD age categories, summary statistics, geographic distribution, LOCP attributable to tobacco in Ireland compared to the EU.

Data were obtained from 2019 GBD line and plot graphs (https://vizhub.healthdata.org/gbd-compare (accessed on 3 April 2022). Advanced settings were applied to obtain data for this research project. Data was selected as follows; cause-specific data were selected, under the following classifications, B 1.1 lip and oral cavity cancer, B 1.2 nasopharynx cancer, B 1.3 other pharynx cancer. Age was set as either age-standardised, 15–49 years, 50–69 years or 70+ years. Gender was set as either male, female or both. Location was set to European Union. Summary measures of deaths, ASIR, ASMR, YLL, YLD and DALYs were selected. Rate was the unit selected. Figure 1 outlines an example of the search strategy.

Once correct variables had been selected, data were inputted into Microsoft Excel sheets. Data were analysed as follows; overall trends, LOCP categories, GBD age categories, summary statistics, geographic distribution, LOCP attributable in Ireland compared to the EU. A decrease from 1990 to 2019 suggested a decline in cases. An increase from 1990 to 2019 suggested a rise in cases. Where relevant, estimated annual percentage changes (EAPC) were calculated. The following calculation was used to calculate EAPC:$$\frac{2019\left(r\right)-1990\left(r\right)}{1990\left(r\right)}$$
r = rate.

A negative EAPC suggested a decrease, whereas a positive EAPC suggested an increase.

## 3. Results

A key finding was the 45% increase in incident cases of LOCP between the years 1990 and 2019. The age-standardised incidence rate (ASIR) of LOCP increases by 3.1%. During the thirty-year period investigated in this study, a 24.7% and 10.1% increase was estimated in the absolute counts of LOCP deaths and DALYs, respectively, whereas the age-standardised mortality (ASMR) and DALY rates decreased by 16% and 21%, respectively (Table 1).

We observed variations in LOCP ASIR when each category was analysed separately. Between 1990 to 2019, the ASIR of lip and oral cavity cancer (LO) decreased from 6.2 new cases per 100,000 (95% UI: 6.1–6.4) in 1990 to 5.3 new cases per 100,000 (95% UI: 4.6–6.1) in 2019, whereas the ASIR of nasopharynx cancer (NP) increased from 1 new case per 100,000 (95% UI: 0.9–1.0) to 1.1 new cases per 100,000 (95% UI: 1–1.3) during the same period. The ASIR of other pharynx cancer (OP) increased from 2.2 new cases per 100,000 (95% UI: 2.2–2.3) in 1990 to 3.3 new cases per 100,000 (95% UI: 2.9–3.8) in 2019 (Figure 2).

Figure 2: ASIR of LOCP between 1990 and 2019.

**Figure 2 ijerph-19-06532-f002:**
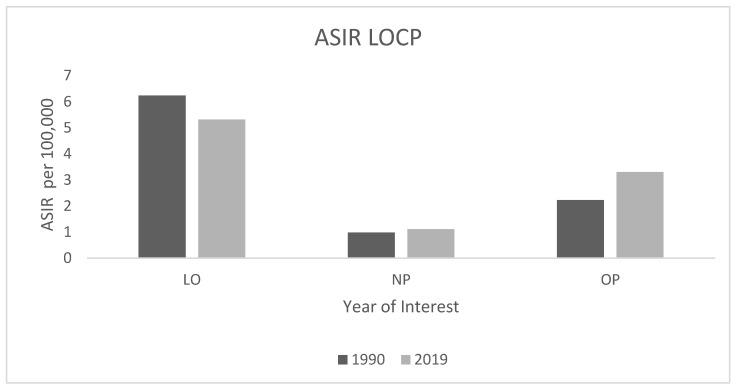
ASIR of LOCP between in 1990 and 2019. LO: lip and oral cavity cancer; NP: nasopharynx cancer; OP: other pharynx cancer.

Between 1990 and 2019, years of life lost (YLL) due to LOCP remained worryingly high, whereas the years lived with disability (YLD) was much lower. The age-standardised LOCP YLLs decreased from 124.8 per 100,000 (95% UI: 122.3–127.3) in 1990 to 97.5 per 100,000 (95% UI: 91.3–104) in 2019. The age-standardised LOCP YLDs increased from 3.8 per 100,000 (95% UI: 3.1–5.1) in 1990 to 4.2 per 100,000 (95% UI: 3–5.6) in 2019 (Figure 3).

Eastern and northern Europe accounted for 34% and 18% of LOCP DALYs in 1990. This estimate increased to 39% and 22%, respectively, in 2019. Western and southern Europe accounted for 27% and 21% of LOCP DALYs in 1990. This estimate decreased to 21% and 18%, respectively, in 2019 (Figure 4).

The age-standardised LOCP DALYS are displayed in Figure 5 and according to UN classification in Table 2, with EAPC DALY calculated for each country.

In Ireland, OP attributable to tobacco decreased from 11.0 DALYS per 100,000 (95% UI: 9.7–12.4) in 1990 to 10.3 DALYS per 100,000 (95% UI: 8.5–12.5) in 2019, corresponding to a 6.4% decrease (Figure 6).

In the EU OP attributable to tobacco decreased from 27.4 DALYS per 100,000 (95% UI: 24.5–30.0) in 1990 to 24.3 DALYS per 100,000 (95% UI: 21.1–27.4) in 2019, corresponding to a 11.3% decrease (Figure 7).

## 4. Discussion

During the thirty-year observation of EU nations, a 24.7% and 10.1% increase was estimated in the absolute counts of LOCP deaths and DALYs, respectively, whereas the age-standardised mortality and DALY rates decreased by 16% and 21%, respectively. This highlights the variations in the underlying age composition of the population, and indicates potential drivers of such an epidemiologic pattern. This was also reflected in the wide 95% uncertainty intervals for ASIR of LOCP (from 9.2–9.7 (1990) to 8.4–11.2 (2019)).

Between 1990 and 2019, YLL due to LOCP remained worryingly high. This suggests that premature mortality is still a public health concern. Notably, the gap between YLL attributable to LO and OP has narrowed considerably since 1990. This possibly suggests that patients with OP are surviving longer compared with those diagnosed with LO.

YLD estimates were proportionately lower compared with YLL estimates in both 1990 and 2019. Based on these findings, mortality is still very high compared with people living with a disability. This suggests that LOCP patients are dying prematurely before their expected life expectancy, whereas low YLD estimates suggest that LOCP survival rates are poor. These observations suggest that slow progress has been made in overall oral healthcare development across EU nations. Countries with high YLLs and YLDs warrant that governments’, researchers’, and policymakers’ work collaboratively to promote effective population-level LOCP preventive strategies to tackle this rising burden of LOCP in the EU.

There were wide variations in all summary metrics in LOCP in European regions, reflecting population-level variations in the distribution of risk factors, such as tobacco and alcohol consumption patterns. These geographic variations in estimates also question the current medical model of oral healthcare in the EU, both in terms of quality and access. Eastern Europe had the highest proportion of LOCP age-standardised DALYs in 1990 and 2019. It was interesting to note that western and southern Europe’s age-standardised LOCP DALYs decreased from 1990 to 2019; however, eastern and northern Europe increased during the same time period. Such geographic variations across EU nations reinforce the need for an integrated oral health care approach-more in alignment with the concept of universal health coverage.

In 1990, France had the highest male LOCP incidence in all age groups. However, in 2019 male LOCP rates ranked eighth for 15–49 year olds, fourth for 50–69 year olds and third for 70+ year olds, suggesting a possible cohort effect. Declining female LOCP trends in France were also observed. Potential reasons for these age-gender variations include improved oral public health care and universal health care [12]. As part of the National Health Strategy 2018–2022, the French government has increased health financing and per-capita health investment by allocating EUR 400 million over five years to support prevention programmes [12]. Although smoking and alcohol consumption still remains above the EU average, stricter preventative legislation and enforcement has shown to be beneficial to date [12].

Hungary had the highest LOCP figures for females in 1990 and 2019 aged 15–49 years old and for males and females aged 50–69 years old in 2019. Likely reasons for this observation include excessive consumption of tobacco and alcohol, along with a poorly structured Hungarian healthcare system [13]. Half of all deaths in Hungary can be attributed to behavioural risk factors, including poor nutrition, high tobacco and alcohol consumption, along with low physical activity [13]. More than one in four adults reported smoking daily in 2014, one of the highest rates in the EU [13]. Smoking rates are more than two times higher among the least educated people [13]. The life expectancy of the Hungarian population remains almost five years below the EU average and substantial inequalities persist [13]. The Hungarian health system remains chronically underfunded, and health does not appear to be a high priority [13]. It is organised around a single health insurance fund and is highly centralised [13]. The shortages and uneven distribution of health professionals also undermines access to health services [13]. Efforts have recently been made by the government, with substantial remuneration raises in an attempt to attract and retain doctors and other health professionals [13].

In contrast, Cyprus showed a consistent pattern across all age groups. For instance, the LOCP incidence rate was lowest in Cyprus for males aged 50–59 years in 1990 and 2019. Evidence suggests that citizens of Cyprus have the highest life expectancy in the EU, and this can be largely attributed to overall good health [14]. Alcohol consumption is low relative to other EU member states and dietary habits are generally favourable signifying overall better health performance [14]. Determining the effectiveness of the health system in Cyprus is challenging due to limited data availability [14]. It is known that, unlike most other countries in the EU, not all Cypriots are covered by the publicly funded health system [13]. Entitlement to free services is largely dependent on annual income levels, with the eligibility threshold varying according to the number of dependents [14].

In Ireland, despite therapeutic approaches remaining constant, successful public health policy along with demographic improvements in population-level oral conditions has resulted in an overall decreased burden of tobacco-related LOCP [15]. LO and NP attributable to tobacco both decreased by 51.7% between 1990 and 2019. This was compared with a 36% LO and 43.7% NP decrease in the EU during the same period. However, a recent plateau was observed for NP, and this must be addressed. It must also be highlighted that OP attributable to tobacco only decreased by 6.4% in Ireland since 1990 in comparison to a 11.3% reduction in the EU (Figure 6 and Figure 7). These results emphasise that Irish tobacco control measures have been successful but that additional preventative policies must be put in place to further reduce the LOCP burden.

## 5. Conclusions

In conclusion, this study highlights the geographic variations in the burden of LOCP across EU nations. Eastern European countries, such as Hungary, have the highest LOCP rates. There has been a rise in nasopharynx and other pharynx cancer rates in recent years, whereas a decline has been noted for lip and oral cavity cancer. Understanding and monitoring the epidemiologic changes to LOCP burden in Europe is important to inform and evaluate public health interventions. Furthermore, the high mortality rate attributable to LOCP is worrying as LOCP is largely preventable. Therefore, a rethinking of the current oral healthcare system in EU nations is necessary to tackle the rising burden of LOCP.

## Figures and Tables

**Figure 1 ijerph-19-06532-f001:**
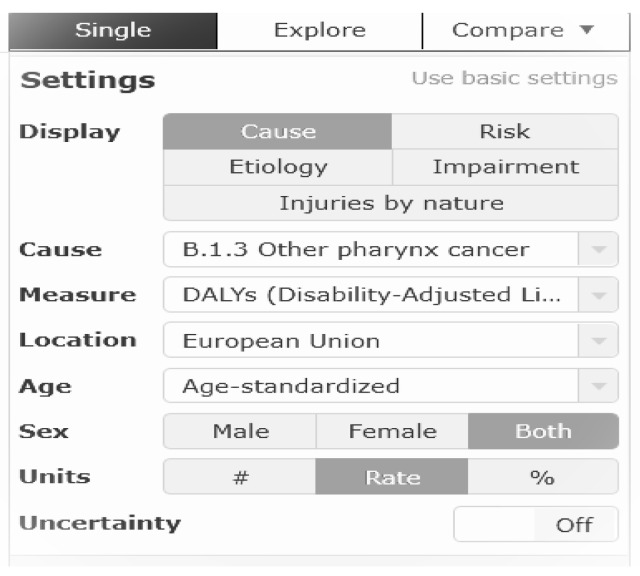
Search strategy employed to obtain relevant Irish LOCP tobacco associated data from IHME database. Source: Institute for Health Metrics Evaluation. Used with permission. All rights reserved [1].

**Figure 3 ijerph-19-06532-f003:**
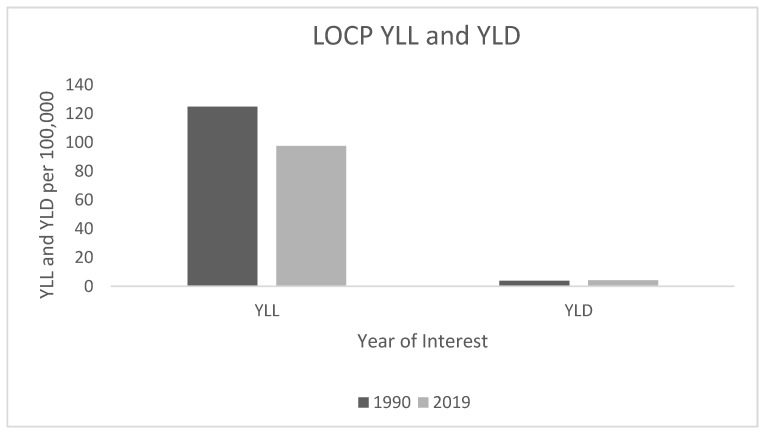
LOCP YLL and YLD trends in 1990 and 2019. YLL: years of life lost; YLD: years lived with disability.

**Figure 4 ijerph-19-06532-f004:**
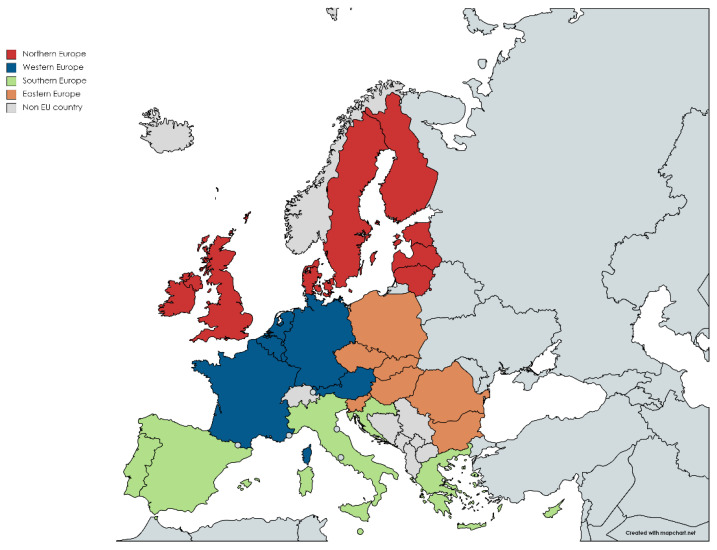
Map of EU DALYs 2019 according to United Nation Geoscheme classification. Source: MapChart.net [10,11].

**Figure 5 ijerph-19-06532-f005:**
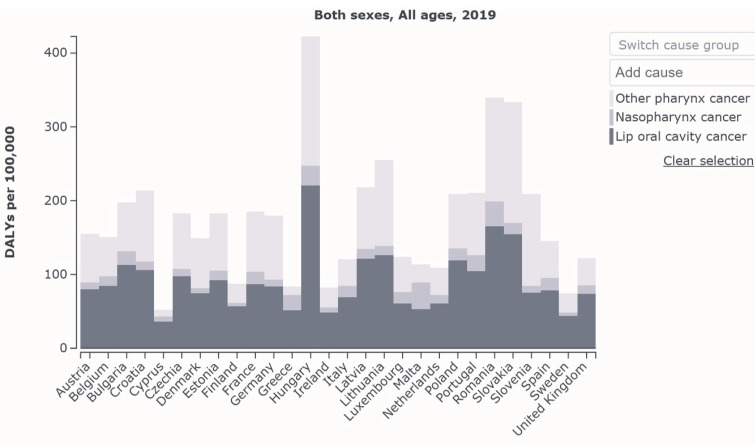
LOCP age-standardised DALYs 2019. Source: Institute for Health Metrics Evaluation. Used with permission. All rights reserved [1].

**Figure 6 ijerph-19-06532-f006:**
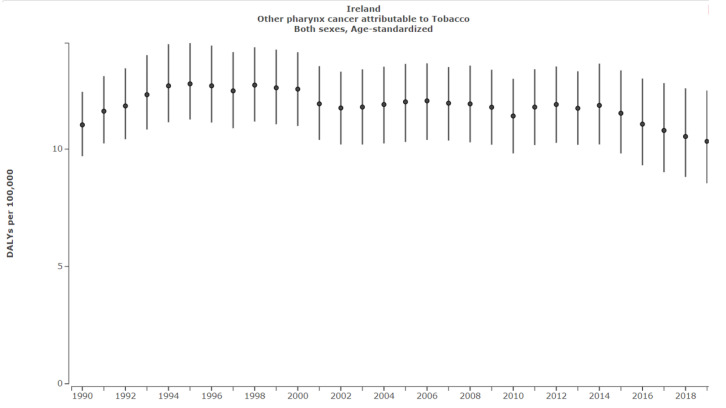
Other pharyngeal cancer in Ireland attributable to tobacco. Source: Institute for Health Metrics Evaluation. Used with permission. All rights reserved [1].

**Figure 7 ijerph-19-06532-f007:**
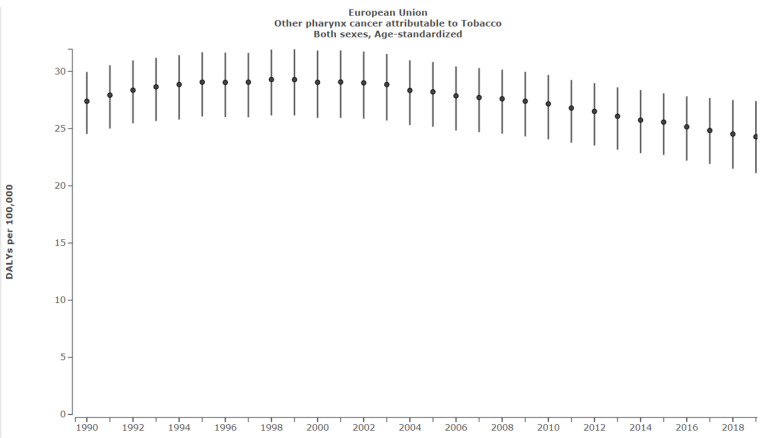
Other pharyngeal cancer in Ireland attributable to tobacco. Source: Institute for Health Metrics Evaluation. Used with permission. All rights reserved [1].

**Table 1 ijerph-19-06532-t001:** Overall Trends of LOCP incidence, mortality rate and DALY in Europe, comparing 1990 and 2019.

	1990	2019	% Change
New cases LOCP (*n*)	59,568.1 (95% UI: 57,880 –61,124)	86,920 (95% UI: 75,588–99,683).	45
ASIR LOCP	9.4 (95% UI: 9.2–9.7)	9.7 (95% UI: 8.4 –11.2)	3.1
Mortality LOCP (*n*)	27,693 (95% UI: 26,961–28,280)	34,545 (95% UI: 32,175–36,740)	24.7
ASMR LOCP	4.3 (95% UI: 4.2–4.4)	3.61 (95% UI: 3.4–3.9)	−16
DALYs LOCP (*n*)	789,831 (95% UI: 772,220–807,837)	869,457 (95% UI: 813,399–930,343)	10.1
Age-standardised DALY rate LOCP	128.7 (95% UI: 125.8–131.6)	101.6 (95% UI: 94.9–108.8)	−21

**Table 2 ijerph-19-06532-t002:** Age-standardised DALYs per 100,000 in 1990 and 2019, with EAPC, countries displayed according to UN European Classification.

Eastern Europe	1990	2019	% DALY Change	Northern Europe	1990	2019	% DALY Change	Southern Europe	1990	2019	% DALY Change	Western Europe	1990	2019	% DALY Change
Slovenia	181.4	121.9	−32.8%	Denmark	53.6	89.2	66.4%	Croatia	206.6	124.7	−39.4%	Netherlands	62.4	62.6	0.3%
Bulgaria	81.6	120.5	47.7%	Latvia	135.6	131	−3.4%	Cyprus	42.3	37.4	−11.6%	France	253.6	115.5	−54.5%
Czechia	131.2	111.7	−14.9%	Lithuania	152.9	196.4	28.4%	Portugal	116.1	126.2	8.7%	Austria	101.9	92.9	−8.8%
Poland	122.5	131.6	7.4%	Ireland	79.4	57.9	−27.1%	Spain	128.8	84.9	−34.1%	Germany	126.5	101.2	−20.0%
Hungary	268.3	262.3	2.2%	United Kingdom	73.2	77.5	5.8%	Malta	92.8	124.7	34.4%	Luxembourg	144.7	83.4	−42.4%
Romania	115.7	216.9	87.5%	Finland	48.2	48.8	1.2%	Italy	108.4	68.9	−36.4%	Belgium	99.6	92.6	−7.0%
Slovakia	276.6	215.7	−22.0%	Sweden	50	42.7	−14.6%	Greece	42.5	43.3	1.9%				
				Estonia	138.6	112.6									
Total	1177.3	1180.7			731.5	756.1			737.5	610.1			788.7	548.2	
Average	168.2	168.7			91.4	94.5			105.4	87.2			131.5	91.4	

## Data Availability

Access to Global Burden of Disease database is available via https://vizhub.healthdata.org/gbd-compare/ (accessed on 3 April 2022).

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
