# Peer review of "Lip, Oral Cavity and Pharyngeal Cancer Burden in the European Union from 1990–2019 Using the 2019 Global Burden of Disease Study"

_ijerph, 2022, doi:10.3390/ijerph19116532_

Round 1

Reviewer 1 Report

I've reviewed the manuscript ijerph-1699964 "Lip, Oral Cavity and Pharyngeal Cancer Burden in European Union using Global Burden of Disease Study 2019" with interest and found it interesting. Please find my comments below. 

  1. Abstract, results presented have too much information and are unclear, please rephrase this and summarise briefly for a better understanding
  2. Line 44 GDB, should be abbreviated earlier the line 42
  3. Authors are encouraged to provide the "aims and objectives" of the present study at the end of the "Introduction" section. 
  4. It would be better if the authors can provide a table or describe with more clarity about YLL and YLD based on DALYs.
  5. Line 54-56, GBD classification of data collection is an important methodology, please provide a tabular or graphic description for better understanding. 
  6. Figures 4-6 are unreadable, please provide a better quality image. 
  7. There are wide variations in all data summary metrics in LOCP, can you please make this data simpler for a better understanding 
  8. Table 2 was discussed in the text properly. Authors should consider re-drawing this table with the comparison between 1990 and 2019 and providing a DALYs change over the years.  

Reviewer 2 Report

The manuscript presents some epidemiological results regarding oral cancer (which includes lip, oral cavity, and pharynx) for the years 1990 and 2019 in Europe. Specifically, the authors analyzed disease burden data to carry out the analysis. The results and the corresponding discussion are vital to assess the present moment and help elaborate health policies that improve the situation. The article is clear and well written, presenting the most pertinent data to the discussion. I have a few suggestions to improve the reading of the article.

In the methods section, the years under study are not referred, although it is mentioned later in the text. My suggestion is that the authors indicate the reason that led to the choice of these years. Authors should also describe what type of statistical analysis was carried out; it is unclear whether they just collected the data and are presenting it in this manuscript or whether they have performed some computation on it.

Figure 3 indicates the necessary information, but it would be more interesting if the same information were presented in the form of a map of Europe with countries colored according to percentages.

Figures 4, 5 and 6 do not have sufficient resolution and are difficult to read. Please replace them.

To have a comparison value, my suggestion is that the authors present a DALY value for, for example, cancer (without specifying the type) or another disease with high impact.

The discussion does not mention whether the therapeutic approaches changed between 1990 and 2019. If so, the authors should indicate this fact and how it may have contributed to the change in burden.
